# Relationship between the Viral Load in Patients with Different COVID-19 Severities and SARS-CoV-2 Variants

**DOI:** 10.3390/microorganisms12030428

**Published:** 2024-02-20

**Authors:** Andrea Santos Coy-Arechavaleta, Julio Elias Alvarado-Yaah, Luis Antonio Uribe-Noguez, Francisco Xavier Guerra-Castillo, Clara Esperanza Santacruz-Tinoco, Eva Ramón-Gallegos, José Esteban Muñoz-Medina, Larissa Fernandes-Matano

**Affiliations:** 1Escuela Nacional de Ciencias Biológicas, Programa de Doctorado en Ciencias en Biomedicina y Biotecnología Molecular, Instituto Politécnico Nacional-IPN, Mexico City 11350, Mexico; santoscoycita@gmail.com (A.S.C.-A.); evaramong@yahoo.com.mx (E.R.-G.); 2Laboratorio Central de Epidemiología, Instituto Mexicano del Seguro Social, Mexico City 02990, Mexico; alvaram25@outlook.com (J.E.A.-Y.); luisuriben@outlook.com (L.A.U.-N.); 3Unidad de Investigación Médica en Inmunología e Infectología “Dr. Daniel Méndez Hernández”, La Raza, Instituto Mexicano del Seguro Social, Mexico City 02990, Mexico; pacoxguerra@gmail.com; 4División de Laboratorios Especializados, Instituto Mexicano del Seguro Social, Mexico City 07760, Mexico; cest03b@gmail.com

**Keywords:** viral load, ddCt, outcome of COVID-19, severity

## Abstract

SARS-CoV-2 has spread throughout the world since 2019, changing in its genome and leading to the appearance of new variants. This gave it different evolutionary advantages, such as greater infectivity and/or a greater ability to avoid the immune response, which could lead to an increased severity of COVID-19 cases. There is no consistent information about the viral load that occurs in infection with the different SARS-CoV-2 variants, hence, in this study we quantify the viral load of more than 16,800 samples taken from the Mexican population with confirmed diagnosis of COVID-19 and we analyze the relation between different demographic and disease variables. We detected that the viral load caused by different variants differs only in the first two days after the onset of symptoms, being higher when infections are caused by the delta variant and lower when caused by omicron. Furthermore, the viral load appears to be higher in outpatients compared to hospitalized patients or in cases of death. On the other hand, no differences were found in the viral load produced in vaccinated and unvaccinated patients, nor did it differ between genders.

## 1. Introduction

In late 2019, SARS-CoV-2 emerged in Wuhan, China. This virus spread worldwide and by the end of 2022 had already caused more than 6 million deaths, in addition to global economic havoc [1].

Since its emergence, SARS-CoV-2 has been mutating, giving rise to variants, which may have characteristics that confer evolutionary advantages over the original Wuhan virus [2], thus causing a current interest to identify variants that may be of concern. For this reason, variants have been classified by the Centers for Disease Control and Prevention (CDC) based on their attributes as a variant under monitoring (VUM), variant of interest (VOI), variant of concern (VOC), and variant of high consequence (VOHC). In addition, the World Health Organization (WHO) convened a group of experts to assign an easy and practical nomenclature to these variants, and they agreed on letters of the Greek alphabet [3]. In Mexico, multiple SARS-CoV-2 variants have been documented throughout this pandemic, including Alpha, Beta, Mu, Lambda, Gamma, Delta and Omicron [4].

Around the world, there are already several studies showing that some of these variants, compared to the original Wuhan variant, have a greater capacity to spread and to evade the immune response, for example, reducing the ability of antibodies, produced by the host, to neutralize the virus and prevent its entry into the cell [5]. Some studies mention that infection with the Delta variant, for example, results in a higher viral load compared to infection with other variants [6,7,8,9]. This could indicate that some variants may have a greater replicative capacity, thus generating more copies of the virus during infection. Viral load, together with the number of antibodies produced by the host, appears to affect infectiousness, although there is still controversy about this assumption in the literature [6,10,11].

In addition, some studies affirm that the severity of COVID-19 is related to a higher viral load of SARS-CoV-2, associating this variable with a higher probability of host death [12,13,14], although there are also contradictory results on the subject [15,16].

Nevertheless, other studies also mention the influence of the time elapsed from the onset of symptoms to the collection of the sample with respect to the viral load being detected [17,18]. For this reason, studies that take this variable into consideration are needed, so that better conclusions on the subject can be reached.

Therefore, viral load appears to be important for issues related to the transmission and prognosis of COVID-19 disease progression and could vary due to different factors. Due to the contradictions currently found in the literature, a retrospective analysis was performed in this study using data from more than 16,800 patients, with the intention of generating new information to have a better understanding of the prognostic value of viral load.

## 2. Materials and Methods

### 2.1. Study Design

A retrospective cross-sectional study was conducted to determine the viral load caused by infection with different variants of SARS-CoV-2 and its relation to clinical outcomes, demographic data and vaccination status of Mexican patients.

For this purpose, SINAVE (National Epidemiological Surveillance System) and SISCEP (Epidemiological Control System) databases were used. Data were initially selected from 25,002 nasopharyngeal exudate samples from patients with COVID-19 confirmed diagnosis, which had also been sequenced by the Mexican Genomic Surveillance Consortium (CoViGen-Mex). We only included the samples that were processed through the same PCR method (Logix Smart COVID-19^®^), with an interval between the onset of symptoms and collection between 1 and 11 days, with all clinical, demographic and laboratory data (Ct value of the SARS-CoV-2 RdRP gene), and with variant identification. From the total remaining (16,984), the viral load of each sample was calculated using the ΔΔCT method, with generation of a standard curve of the SARS-CoV-2 RdRP gene. Thereafter, 104 samples with outlier ΔCT values were identified using the interquartile range (IQR) (samples with the CT value 3 times the interquartile range value below quartile 1 or above quartile 3). Because these samples contained a CT value that was numerically distant from the rest of the data, which could lead to misleading results, we decided to exclude these samples, leaving a total of 16,880 included for the analysis carried out in this study. Figure 1 shows the summary of how the samples were selected and the associated data (Figure 1).

The information in these databases comes from all over the Mexican territory and was received at the four Epidemiological Surveillance Laboratories of the Mexican Institute of Social Security (IMSS) between 1 March 2021, and 4 September 2022. The methodology used for diagnosis (carried out by the Central Epidemiology Laboratory, IMSS), viral load determination (by generating a standard curve made specifically for this project), and sequencing (made by CoViGen-Mex) was the same for all the samples and is described next [19].

### 2.2. Identification of Positive Cases

For all the samples analyzed in this study, the RNA was obtained from a 140 μL pharyngeal exudate sample taken with a Dacron swab (Copan Diagnostics, Corona, CA, USA, Catalog: 159C) and stored in a viral transport medium (BD™ Universal Viral Transport System, East Rutherford, NJ, USA. Catalog: 220220), with the QIAamp 96 Viral RNA kit (QI-AGEN, Hilden, Germany) used according to the manufacturer’s instructions; the Logix Smart COVID-19^®^ amplification kit (COVID-K-001; Salt Lake City, UT, USA) in the 7500 Fast Real-Time PCR System (Applied Biosystems^®^, Foster City, CA, USA) was used for the diagnosis of COVID-19. This kit detects the RdRP (which is a specific gene for SARS-CoV-2) and RNaseP (RP) genes (which is an endogenous gene of human epithelial cells and was used as an internal control). The genes were evaluated by adding 5 μL of the RNA in a multiplex master reaction of the aforementioned kit, and with the following thermo-cycling conditions: one cycle at 45 °C for 15 min and 95 °C for 2 min and 50 cycles at 95 °C for 3 sec and 55 °C for 32 s. A positive result implied that both genes were detected with Ct values below 37. Samples with CT values greater than 30 and less than 37 were excluded from this protocol as the CoViGen-Mex only sequences samples with CT values below 30. 

-
*Sequencing of the SARS-CoV-2 genome (CoViGen-Mex)*


Sequencing was carried out using RNA leftovers from samples processed by the LAVEs that tested positive with the aforementioned methodology for the COVID-19 diagnosis confirmation. All assays were performed by the Mexican Consortium for Genomic Surveillance (CoViGenMex) using the Illumina NextSeq500 or Miniseq platforms, with cells where 500 or 200 samples were sequenced in parallel, respectively.

The data was then analyzed through a computational pipeline analysis designed by CoViGenMex. The viral sequences, once their quality was confirmed, were released to an international platform Global Initiative on Sharing All Influenza Data for SARS-CoV-2 and influenza sequences, (GISAID www.gisaid.org (accessed on 26 January 2024)), and to the national database, MexCoV2 (http://mexcov2.ibt.unam.mx:8080/COVID-TRACKER/ (accessed on 26 January 2024)). During the analysis of the obtained sequences, mutations were identified. These mutations indicated the lineage of the different SARS-CoV-2 variants that circulated in Mexico, and its results, were deposited in the SINAVE and SISCEP platforms that we used in this work.

### 2.3. Quantification of SARS-CoV-2 Viral Load

-
*Validation of the endogenous control*


With the intention of comparing the viral load generated by the different variants of SARS-CoV-2, a relative quantification analysis was designed using the human RP gene as an endogenous gene. First, it was validated that the expression levels of the reference gene (RP) were similar to those of the problem gene of RNA-dependent RNA polymerase (RdRP).

A concentrated solution of 10^10^ theoretical copies were prepared for the problem gene and the endogenous gene, taking as reference the weight of a base pair of 650 Da, and a size of 200 bp for RdRP and 180 bp for RP. Subsequently, the number of molecules per microgram was calculated using Avogadro’s number and the number of moles [20].

From the concentrated solutions, in each case, decimal dilutions were made to generate a 10-point standard curve (10^10^–100 copies/uL). The slope of the curve generated by plotting the ΔCT obtained from the CT values for the different dilutions was between −0.1 and 0.1, so the RP gene could be used as an endogenous control in this case [21].

The 2^−ΔΔCT^ method, better known as ΔΔCT, is a relative quantification strategy for the results of a qPCR or RT-qPCR, which uses the generated threshold cycle (CT), assuming an amplification efficiency of 100% in the analyzed samples. The two “deltas” present in the name of this method refer to the fact that the expression level of a target sample is compared to a control or reference sample, also using a reference gene as a normalizer. The results of this method are usually reported in increments from one sample to the other; however, in this work, we only subtract the CT of the endogenous gene (RP) from the CT of amplification of the RdRP gene of SARS-CoV-2. Subsequently, using the resulting CT of each sample (ΔCT), we analyzed the means for comparisons between groups. The ΔCT values shown are inversely proportional to the SARS-CoV-2 viral load.

### 2.4. Statistical Analysis

Descriptive statistics were used to report averages and percentages; these were calculated with their respective 95% confidence intervals. Outliers were identified using the IQR (interquartile range). The chi-square test was used to compare categorical variables. To cross-reference the factors with the numerical variables, the one-way parametric ANOVA test or the nonparametric Kruskal-Wallis H test was used, as appropriate. *p* values < 0.05 were considered significant. RStudio (version 2023.06.1+524, Boston, MA, USA) and Microsoft Excel (version 16.66.1, Redmond, WA, USA) were used for the analysis and generation of graphs.

## 3. Results

Of the 16,984 samples that went through the viral load quantification process, only 16,880 were analyzed, since 104 were identified as outliers (analysis carried out by the interquartile range (IQR) method). 

The analysis of the time between the onset of symptoms and collection of the sample, which we defined as the timing of sample collection, showed that the viral load decreased significantly over the days (*p* < 0.05, Figure 2); therefore, the total samples analyzed were divided according to this parameter, even though most of the analyzed data came from patients who sought medical attention between 1 and 4 days after the symptoms appeared (Group A and B), as shown in Table 1.

The Delta and Omicron variants were detected most frequently during the analysis period, with prevalences of 44.2 and 49.0%, respectively. The Alpha and Gamma variants circulated 21.4 and 12.9 times less than Omicron, and the least frequent were Beta, Lamb-da and Mu, which together were detected only in 115 (0.7% of the total). Therefore, in this work, and for ease of analysis, we combined them and refer to them as “Others” (Table 1).

In Table 1, we also include other data, such as those referring to sex, age, status of vaccination and type of patient (outpatient, hospitalized or death) at the time of collection of the sample.

When analyzing the data, we observed that only at the beginning of the infection (Group A) the viral load produced at the time of collecting the samples is different between the variants (Figure 3). We found that individuals infected with Gamma had a significantly higher viral load compared to those infected with Omicron and Alpha (*p* < 0.05, Figure 3A), as well as those infected with Delta in relation to those infected with Omicron (*p* < 0.05, Figure 3A). After 3 days the onset of symptoms appeared, no difference in the viral load produced during infection was detected between the different variants, as shown in Figure 3B–D.

Another analysis carried out was that of the viral load with respect to the sex of the patient. However, no statistically significant differences were found in any of the Groups, as shown in Appendix A.

On the other hand, the SARS-CoV-2 viral load seemed to be related to the age of the participants; however, as in the analysis of the variants, we detected differences only at the beginning of the infection, in this case, up to 4 days after the onset of symptoms (Group A and B).

As shown in Figure 4, older adults had a lower viral load compared to the other age groups (Group A: 0–9: −5.0734, 10–19: −5.6155; 20–59: −5.3863; ≥60: −4.6224; *p* < 0.05) (Group B: 0–9: −4.7212, 10–19: −4.8003; 20–59: −4.4685; ≥60: −3.9526; *p* < 0.05), and the differences were significant only in comparison with the groups of 10–19 and 20–59 years, and only during the first four days after the onset of symptoms.

The period of time included in this study covered the beginning of the vaccination campaigns in Mexico in which, at that time, contemplated only adults, prioritizing the immunization of older adults; therefore, vaccination status could be affecting the results. However, even if this variable was taken into consideration, no significant differences were found (Appendix A).

Another variable considered in this work was the severity of COVID-19, due to its possible relation with the viral load of SARS-CoV-2. Contrary to expectations and, regardless of the time of collecting the sample, outpatients always showed a higher viral load than hospitalized patients and/or those who had died (Figure 5). This analysis is performed in a general manner, but it should be noticed that the vast majority of hospitalized and dead patients were older adults, as shown in Table 2.

Still on this topic, we decided to also analyze the different variants regarding patient severity. The results showed that, unlike viral load, which is only related to variants at the onset of infection, severity status appears to depend on variants at all stages of the disease (Figure 6 and Figure 7). 

According to the correspondence analysis with dimension reduction (Figure 7), we can see that the Delta variant is the most related to death cases. When dichotomizing Delta with respect to the other variants, the differences were significant in all groups (Group A: 10.2% vs. 2.4%, *p* < 0.05; Group B: 14.7% vs. 6.3%, *p* < 0.05; Group C: 23.8% vs. 15.2%, *p* < 0.05; Group D: 28.9% vs. 23.1%, *p* < 0.05; and independent of OPT: 18.4% vs. 6.0%, *p* < 0.05). On the other hand, this same analysis showed that the Omicron variant is the most related to outpatient cases and, when dichotomizing this variant with respect to the others, the differences were significant except in Group D (Group A: 83.7% vs. 72.4%, *p* < 0.05; Group B: 74.6% vs. 62.7%, *p* < 0.05; Group C: 52.2% vs. 45.1%, *p* < 0.05; Group D: 22.3% vs. 27.0%, *p* > 0.05; and independent of OPT: 76.3% vs. 53.9%, *p* < 0.05). 

In addition, analyses were performed to further elucidate the relation between viral load, severity level, and different variants. The results showed that the viral load of those infected with the Delta variant was lower in severe cases compared to outpatient cases, throughout the evolution of the disease (Figure 8). On the other hand, this pattern could not be detected in patients infected with other variants, except in groups B and D in those infected with the Gamma variant, and in group A in those infected with Omicron.

## 4. Discussion

This is the first study that evaluates SARS-CoV-2 viral load distribution in so many epidemiological data in a large number of patient samples in Mexico. A total of 16,880 samples were analyzed to investigate the relation among vaccination status, patient outcomes, age, sex, and SARS-CoV-2 variant with respect to the viral load produced during infection, which was determined by using the ΔCT method, normalizing with the CT value of the housekeeping RP gene and, thus, eliminating the variability due to sampling as mentioned in other articles [10].

According to the obtained results, the viral load was significantly higher at the beginning of the infection and decreased over time. Other studies also showed similar results, with the viral load decreasing as the disease progressed [16,22,23,24], even though the sample collection and diagnostic method were different. For this reason, all analyses in this study were performed independently for the groups formed with respect to the time between the onset of symptoms and collection of the sample (A, B, C, and D).

One of the main concerns that we tried to resolve in this work was the possibility that the mutations that gave rise to the new SARS-CoV-2 variants could affect the speed of virus replication in patients, causing a higher viral load during infection and consequently a worse outcome of the disease. Regarding this issue, our results showed differences between the viral loads of the different variants only at the beginning of the infection (Group A), when the Delta and Gamma variants seemed to produce a higher number of copies than the Alpha and Omicron variants. Similarly, in a study carried out by Puhach and collaborators [23], a higher viral load was found in Delta than in Omicron at the beginning of the infection. Other studies also suggested that the viral load in patients infected with the Delta variant was higher than in those infected with the Omicron variant [6] or found a higher viral load with Delta variant infection compared to the Alpha variant [25]. However, some studies did not detect any difference, such as that of Yuasa and collaborators [26], where they analyzed 694 nasopharyngeal exudates through RT-qPCR and sequencing and reported the number of copies produced by the Delta variant vs the Omicron variant. Other studies found that the Delta variant presented a viral load ten times higher than the historical variants and a significantly higher difference with the Beta variant, but no statistical difference was found between the Delta and the Alpha [7].

We also compared the different COVID 19 variants with the patient’s severity; we found statistical differences between Delta and Omicron with the other variants, in which infection with Delta was associated with dead patients, and Omicron was associated with outpatients. Other studies found similar results; one of them showed that during the outbreak of the SARS-CoV-2 Delta variant in Vietnam, the case fatality rate was higher [27]. Another study which included 87 pediatric cases, infected with Alpha (5.7%), Delta (60.9%), and Omicron (33.3%) variants, showed that severe disease, distress, and myalgia were more frequent in Delta-infected patients [28]. Zachary and collaborators studied 102315 confirmed COVID-19 cases, of which 20, 770 were infected with the Delta variant, 52, 605 with the Omicron B.1.1.529 variant, and 28, 940 with the Omicron BA.2 subvariants. Mortality rates were 0.7% for Delta (B.1.617.2), 0.4% for Omicron (B.1.1.529), and 0.3% for Omicron (BA.2). Finally, they concluded that the Omicron BA.2 subvariants were significantly less severe than that of the Omicron and Delta variants [29]. Also, in a study that compared inflammatory markers among patients hospitalized during Omicron infection with those of Alpha and Delta, showed that levels of CRP in Delta and Alpha were significantly higher compared to Omicron; the same trend was observed for ferritin, alanine aminotransferase, aspartate aminotransferase, lactate dehydrogenase, and albumin. So, in accordance with our results, the mortality in Delta and Alpha was higher than Omicron [30].

On the other hand, our study showed that there were differences in the viral load detected in the different types of patients (outpatients, hospitalized patients and deceased patients) in all groups; that is, these differences occurred independently of the timing of sample collection. Interestingly, outpatients always had a higher viral load than hospitalized patients and those who died from the disease. In the literature, some studies propose that the viral load in hospitalized patients may be lower because the severity of the disease is due to factors such as coinfections associated with SARS-CoV-2. Garay and collaborators [31] estimated the association of bacterial pneumonia with the mortality of patients due to COVID-19 and found that 89 of 252 patients tested positive for bacteria that cause pneumonia, which increased the percentage of deaths and could explain why hospitalized patients may have a low viral load but greater severity due to secondary infection.

However, unlike our results, Tsukagoshi and collaborators [30] found that the viral load in deceased patients was significantly higher; although, we must consider that in their study, the sample was composed of only 286 individuals. Similarly, Liu et al. also concluded that patients with severe COVID-19 tend to have a high viral load and a long virus shedding period [31]. Contradictions in results found in the literature may be due to differences in study design and timing of sample collection. Another example of the diversity of published results and conclusions is that some studies have reported that viral load is not related to patient outcome [15,18] or that it is independently correlated with the risk of hospital mortality [32]. Killingley and collaborators [16] conducted a controlled study, inoculating 36 people who had not been in contact with the virus or those previously vaccinated, concluding that there is no relationship between viral load and patient outcome. Although these authors were able to control for many of the confounding factors that could affect the results of their study, their n was very small, so we emphasize the need to conduct larger scale studies to shed more light on this issue.

Regarding the viral load that develops in different age groups, Pu-hach and collaborators [23], did not find significant differences in their review; however, they only compared children and adults. Many studies also found no differences between groups of different ages [32,33,34,35], although it must be taken into consideration that they were smaller studies, that the sampling techniques were not homogeneous, and that they all included a small number of pediatric patients. Aranha and collaborators [25] mentioned that the virus elimination time does not depend on the patient’s age. The results obtained in the aforementioned studies are not consistent with our results (Groups A and B). According to our results, a cross-sectional study in Ghana including 9549 positive samples showed the lowest median viral loads in those aged 10 years and the highest in those aged 71–80 years [36]; we also detected differences between adolescents and adults with respect to older adults, who showed a lower viral load. On the contrary, there was also another article that showed increasing SARS-CoV-2 viral load with increasing age, especially showing low viral loads in children under 12 years of age [37], but they used different sampling methods and did not used the ΔCT method to measure viral load.

One factor that was thought to affect viral load was vaccination. However, in general, in our analysis, we did not find differences in viral load between vaccinated and unvaccinated people, which agrees with the findings of works such as those of Levine and collaborators [38] and Singanayagam and collaborators [39]. The latter demonstrated that although vaccination does not generate changes in the viral load, it does reduce infections by the Delta variant and accelerates the elimination of this virus. On the other hand, in another study carried out on the Delta variant, there was a drastic decrease in the viral load in vaccinated people [23].

Although in this study we analyzed a large amount of data, some variants such as Delta and Omicron had a lot of data; however, some others such as alpha, beta, lambda or mu did not have enough data. The same thing happened with the age groups; the groups from 20 to 59 or over 60 had much more data than the groups from 0 to 9, or from 10 to 19, so it would have been interesting to have a greater number of data from the groups mentioned above. Also, there are a large number of variables that can influence the viral load, although in this work we take into account a large number of variables, we do not include the symptoms or comorbidities of each patient.

## 5. Conclusions

In this study we detected great differences in the viral load depending on the disease evolution (regardless of variable type) and, although the patient’s outcome may vary depending on the SARS-CoV-2 variable contracted, the viral load produced between the variables only differs at the beginning of the disease and appears to be unrelated to age, sex, or vaccination status. Contrary to what was expected, the viral load is much lower in hospitalized patients, including deaths compared to outpatients. The beliefs are that there must be immunological factors as well as coinfections that aggravate the disease condition that should be further investigated.

## Figures and Tables

**Figure 1 microorganisms-12-00428-f001:**
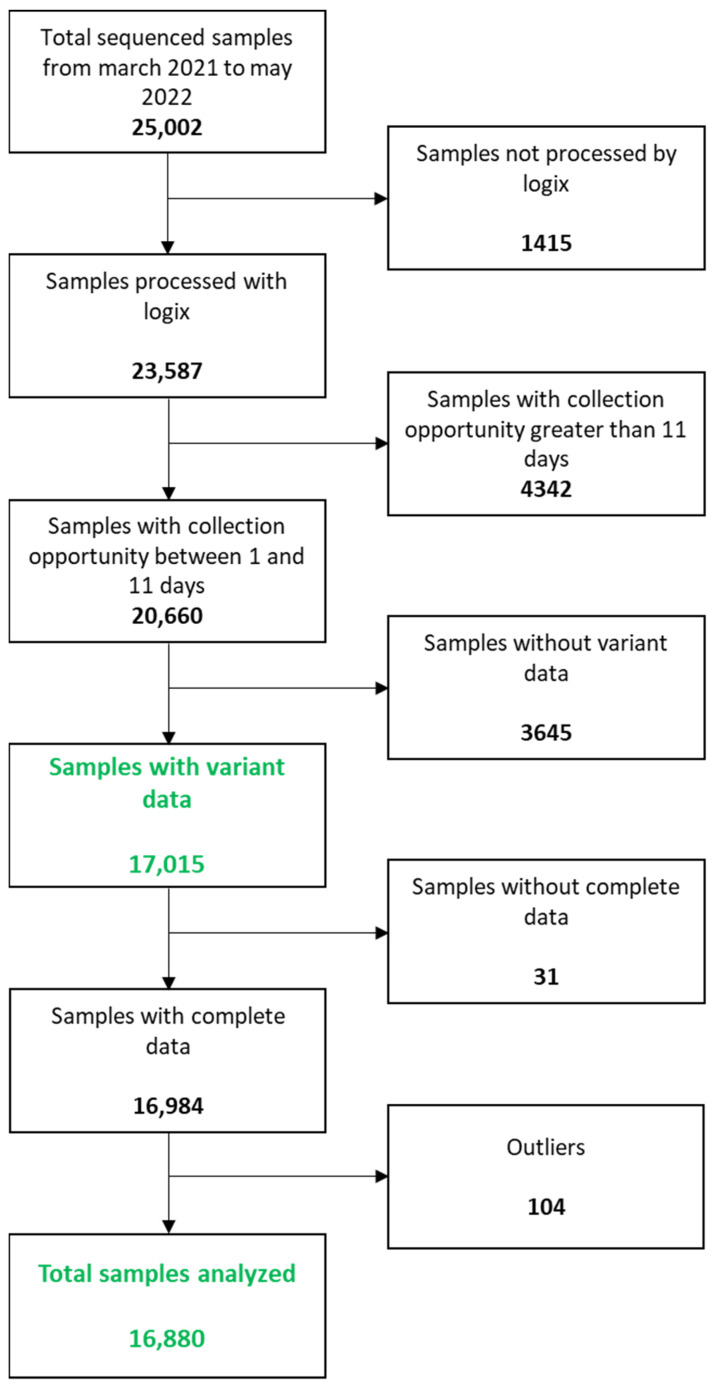
Database curation diagram.

**Figure 2 microorganisms-12-00428-f002:**
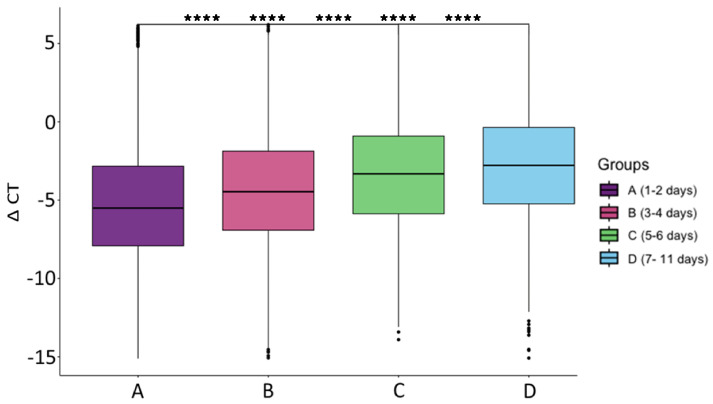
ΔCT mean with respect to the timing of sample collection. A, B, C, D = Group A, B, C and D, respectively. **** *p* < 0.00005.

**Figure 3 microorganisms-12-00428-f003:**
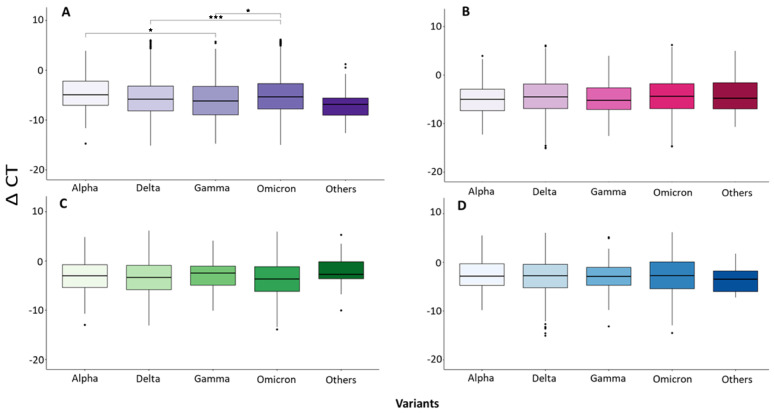
Analysis of the ΔCT detected in patients infected with the different variants. (**A**–**D**) = Group A, B, C and D, respectively. * *p* < 0.05, *** *p* < 0.0005.

**Figure 4 microorganisms-12-00428-f004:**
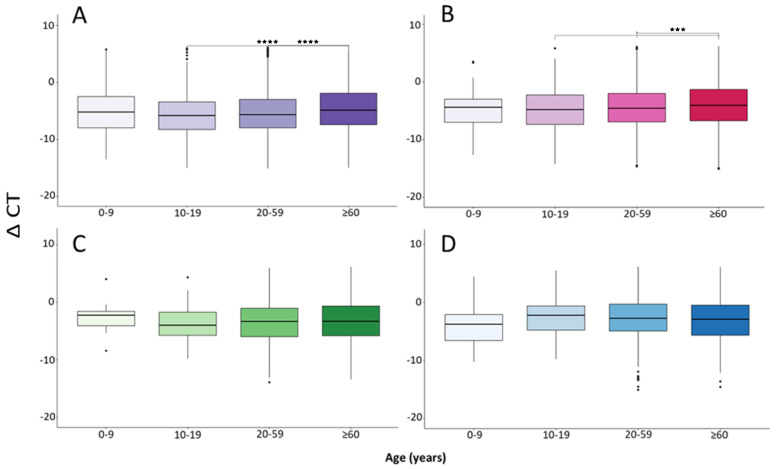
Analysis of the ΔCT detected in the different age groups. (**A**–**D**) = Group A, B, C and D, respectively. *** *p* < 0.0005, **** *p* < 0.00005.

**Figure 5 microorganisms-12-00428-f005:**
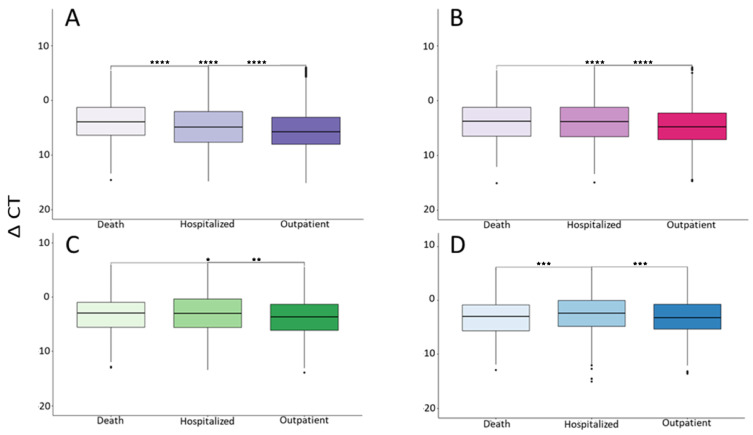
Influence of ΔCT on the severity of COVID-19. (**A**–**D**) = Group A, B, C and D, respectively. * *p* < 0.05, ** *p* < 0.005, *** *p* < 0.0005, **** *p* < 0.00005.

**Figure 6 microorganisms-12-00428-f006:**
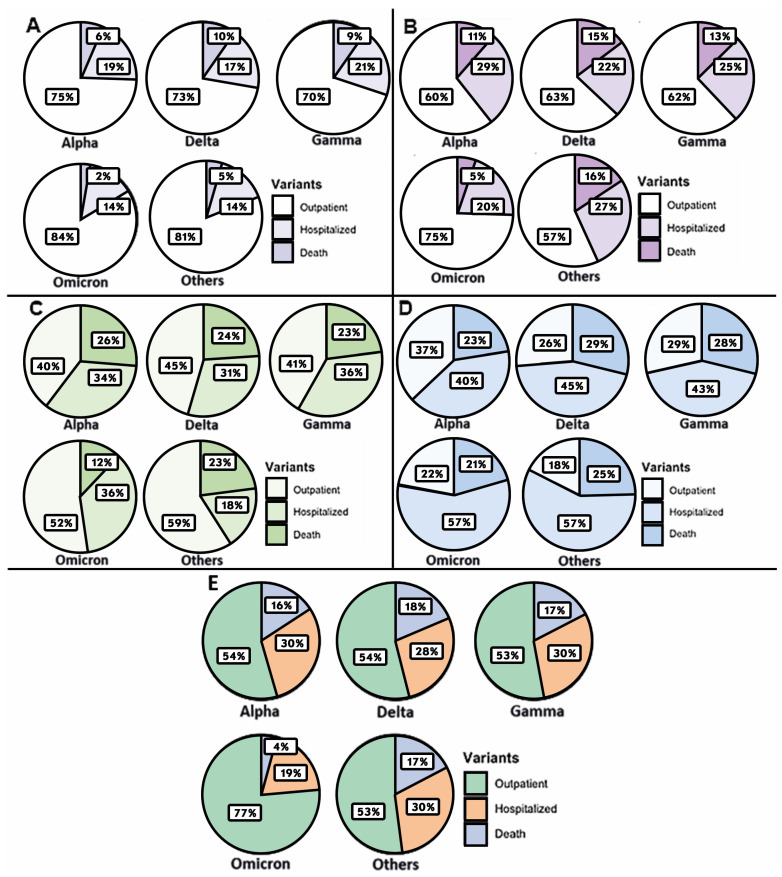
Relation of each variant with respect to COVID-19 severity. (**A**–**D**) results obtained for group A, B, C and D, respectively. (**E**) results obtained for all the data analyzed in the study.

**Figure 7 microorganisms-12-00428-f007:**
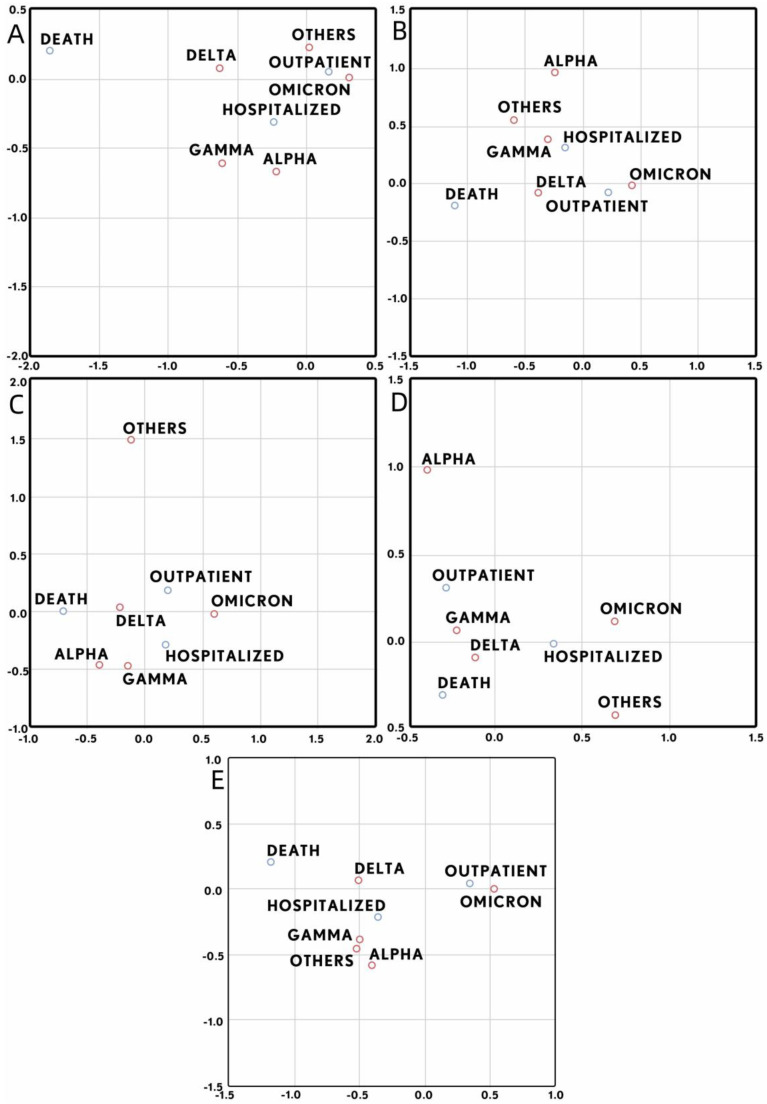
Correspondence analysis with dimension reduction of the relationship between infection with different variants and the severity developed by the host. (**A**–**D**) results obtained for group A, B, C and D, respectively. (**E**) results obtained for all the data analyzed in the study.

**Figure 8 microorganisms-12-00428-f008:**
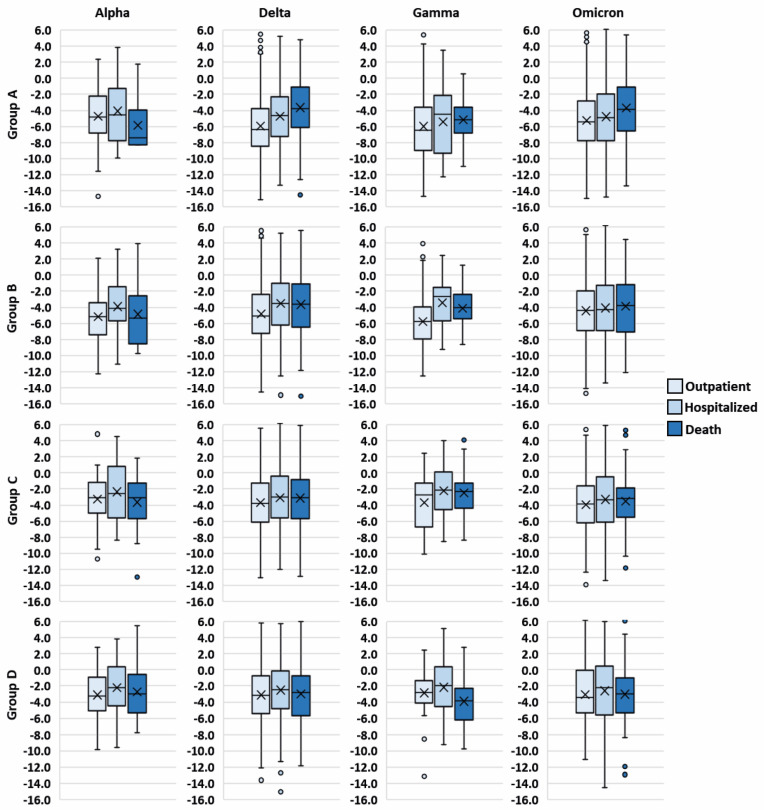
Relationship between ΔCT and disease severity in patients infected with the different variants.

**Table 1 microorganisms-12-00428-t001:** Demographic, clinical and type of patient data.

	Total	Group A (1–2 Days)	Group B (3–4 Days)	Group C (5–6 Days)	Group D (7–11 Days)
**Variants**					
Alpha	386	118	90	76	102
Delta	7463	2224	2137	1312	1790
Gamma	642	213	163	101	165
Omicron	8274	5094	2242	557	381
Others *	115	21	44	22	28
**Age group (years)**					
0–9	457	304	107	21	25
10–19	753	399	233	64	57
20–59	11,609	5768	3239	1258	1344
>60	4061	1199	1097	725	1040
**Vaccines**					
No	10,911	4770	2981	1382	1778
Yes (1 shot)	1651	653	498	233	267
Yes (2 shots)	4318	2247	1197	453	421
**Sex**					
Female	9204	4439	2594	1021	1150
Male	7676	3231	2082	1047	1316
**Patient type**					
Outpatient	10,947	6128	3198	973	648
Hospitalized	4000	1184	1003	668	1145
Death	1931	358	475	427	671

* Beta, Lambda and Mu variants.

**Table 2 microorganisms-12-00428-t002:** Severity status of the participants of each age group.

Age Groups (Years)	Outpatient (%)	Hospitalized (%)	Death (%)	Total
0–9	296 (64.8)	156 (34.1)	5 (1.1)	457
10–19	626 (83.1)	123 (16.4)	4 (0.5)	753
20–59	9075 (78.2)	1876 (16.1)	658 (5.7)	11,609
>60	950 (23.4)	1845 (45.4)	1266 (31.2)	4061

## Data Availability

Data are contained within the article.

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
