# Peer review of "Relationship between the Viral Load in Patients with Different COVID-19 Severities and SARS-CoV-2 Variants"

_microorganisms, 2024, doi:10.3390/microorganisms12030428_

Round 1
Reviewer 1 Report
Comments and Suggestions for Authors
Coy-Arechavaleta et al. have assessed associations of SARS-CoV-2 viral loads with clinical and epidemiological parameters based on a large Mexican dataset. Although the experience is interesting, I have a few recommendations on how the work could be further improved.
1.) Although I am not a native English speaker myself, partly very long sentences (both in the abstract and in the main text) make this manuscript difficult to read. The authors should consider splitting some sentences to make the manuscript text more idiomatic.
2.) Methods chapter, study design and figure 1) The authors should explain more explicitly what they mean with the term “outliers” (line 89 and figure 1).
3.) Methods chapter, line 160: Not all readers will be well familiar with the double-delta-Ct method for quantification purposes. Although the authors provide a reference on this method in the discussion, a more detailed explanation of the approach would be helpful here.
4.) I strongly recommend for the implementation of a limitations paragraph at the end of the discussion, in which the authors summarize the limitations of the interpretability of their results to the readers. At the very least, the COVID-19 associated uncertainties regarding infection timepoint and symptom-onset should be mentioned, as the associated variability can negatively interfere with the estimations of infection stages.
5.) Discussion, lines 346-350: It seems that two sentences have been erroneously merged here.
Author Response
For research article
Relationship between the viral load in patients with different COVID-19 severities and SARS-CoV-2 variants
Response to Reviewer 1 Comments
|
||||||
1. Summary |
|
|
||||
Thank you very much for taking the time to review this manuscript. Please find the detailed responses below and the corresponding revisions/corrections highlighted/in track changes in the re-submitted files.
|
||||||
2. Questions for General Evaluation |
Reviewer’s Evaluation |
Response and Revisions |
||||
Does the introduction provide sufficient background and include all relevant references? |
Can be improved |
The responses were given in the comments. |
||||
Are all the cited references relevant to the research? |
Yes |
|
||||
Is the research design appropriate? |
Yes |
|
||||
Are the methods adequately described? |
Can be improved |
The responses were given in the comments. |
||||
Are the results clearly presented? |
Yes |
|
||||
Are the conclusions supported by the results? |
Can be improved |
The responses were given in the comments. |
||||
3. Point-by-point response to Comments and Suggestions for Authors |
||||||
Comments 1: Although I am not a native English speaker myself, partly very long sentences (both in the abstract and in the main text) make this manuscript difficult to read. The authors should consider splitting some sentences to make the manuscript text more idiomatic. |
||||||
Response 1: We adjusted the text in several parts in a general way, according to the reviewer's suggestion. |
||||||
Comments 2: Methods chapter, study design and figure 1) The authors should explain more explicitly what they mean with the term “outliers” (line 89 and figure 1). |
||||||
Response 2: The text was modified to include more detailed information about outliers excluded from the study: “Thereafter, 104 samples with outlier Ct values were identified using the interquartile range (IQR) (samples with the Ct value 3 times the interquartile range value below quartile 1 or above quartile 3). Because these samples contained a Ct value that was numerically distant from the rest of the data, which could lead to misleading results, we decided to exclude these samples, leaving a total of 16,880 included for the analysis carried out in this study.” |
||||||
|
||||||
Comments 5: Discussion, lines 346-350: It seems that two sentences have been erroneously merged here. |
||||||
Response 5: The paragraph was rewritten: “… although, we must consider that in their study, the sample was composed of only 286 individuals. Similarly, Liu et al also concluded that patients with severe COVID-19 tend to have a high viral load and a long virus shedding period [31]. Contradictions in results found in the literature may be due to differences in study design and timing of sample collection. Another example of the diversity of published results and conclusions is that some studies have reported that viral load is not related to patient outcome [15,18] or that it is independently correlated with the risk of hospital mortality [32]. Killingley and collaborators [16] conducted a controlled study, inoculating 36 people who had not been in contact with the virus or previously vaccinated, concluding that there is no relationship between viral load and patient outcome. Although these authors were able to control for many of the confounding factors that could affect the results of their study, their n was very small, so we emphasize the need to conduct larger scale studies to shed more light on this issue.” |

Reviewer 2 Report
Comments and Suggestions for Authors
More than 16,800 samples taken from the Mexican population with the confirmed COVID-19 cases were studied and the relationship between the viral load of these samples and different demographic and disease variables were analyzed in this article.
Several suggestions
1. Line 88, please define briefly about [ddCt].
2. [RNaseP (RP)] not [RP] in line 114. [RP] not [RNaseP (RP)] in line 146.
3. Lines 151 and 157, [1010] or [1010]?
4. Please add a reference after lines 154 and 164.
5. Line 214, please check [not significant differences were found in none of the Groups].
6. Line 295, [different PCR]?
7. Line 296, [vaccination time]?
8. Line 315, it is better to use [spike] than [S].
9. Line 315, [higher transmission] or [higher replication]?
10. If the results regarding anti-SARS-CoV2 therapy are available in these subjects, it is better to include the relationship between anti-viral effect and viral load, viral variants in this study.
Comments on the Quality of English Language
English editing is suggested: [a] The paragraph from lines 47 to 54 is similar to that from 55 to 60; [b] Lines from 225 to 228; [c] Lines from 346 to 349.
Author Response
For research article
Relationship between the viral load in patients with different COVID-19 severities and SARS-CoV-2 variants
Response to Reviewer 2 Comments
|
||
1. Summary |
|
|
Thank you very much for taking the time to review this manuscript. Please find the detailed responses below and the corresponding revisions/corrections highlighted/in track changes in the re-submitted files.
|
||
2. Questions for General Evaluation |
Reviewer’s Evaluation |
Response and Revisions |
Does the introduction provide sufficient background and include all relevant references? |
Yes |
|
Are all the cited references relevant to the research? |
Yes |
|
Is the research design appropriate? |
Yes |
|
Are the methods adequately described? |
Can be improved |
The responses were given in the comments. |
Are the results clearly presented? |
Can be improved |
The responses were given in the comments. |
Are the conclusions supported by the results? |
Yes |
|
3. Point-by-point response to Comments and Suggestions for Authors |
||
Comments 1: Line 88, please define briefly about [ddCt]. |
||
Response 1: As we decided to use ΔCT averages and never reported the increases between groups (ΔΔCT), we decided to modify the nomenclature of the Y axes of the images. With these modifications, the description remains more faithful to what we analyzed. We added a text in the methodology to give more detailed information about the relative quantification method used and how we report the data obtained: “The 2-ΔΔCT method, better known as ΔΔCT, is a relative quantification strategy for the results of a qPCR or RT-qPCR, which uses the generated threshold cycle (CT), assuming an amplification efficiency of 100% in the analyzed samples. The two "deltas" present in the name of this method refer to the fact that the expression level of a target sample is compared to a control or reference sample, also using a reference gene as a normalizer. The results of this method are usually reported in increments from one sample to the other; however, in this work, we only subtract the CT of the endogenous gene (RP) from the CT of amplification of the RdRP gene of SARS-CoV-2. Subsequently, using the resulting CT of each sample (ΔCT), we analyzed the means for the comparisons between the groups. The ΔCT values shown are inversely proportional to the SARS-CoV-2 viral load." |
||
Comments 2: [RNaseP (RP)] not [RP] in line 114. [RP] not [RNaseP (RP)] in line 146. |
||
Response 2: The changes that were requested have been made. |
||
Comments 3: Lines 151 and 157, [1010] or [1010]? |
||
Response 3: Following the reviewer's observation, the value “1010” was corrected to “1010”. |
||
Comments 4: Please add a reference after lines 154 and 164. |
||
Response 4: We added the requested references. |
||
Comments 5: Line 214, please check [not significant differences were found in none of the Groups]. |
||
Response 5: The text of the mentioned line has been modified: “However, no statistically significant differences were found in any of the Groups, as shown in Figure S1.” |
||
Comments 6: Line 295, [different PCR]? |
||
Response 6: The correction has been made based on the observation of the reviewer: “Other studies also showed similar results, with the viral load decreasing as the disease progressed [16,20–22], even though the sample collection and diagnostic method were different.” |
||
Comments 7: Line 296, [vaccination time]? |
||
Response 7: The correction has been made based on the observation of the reviewer. “For this reason, all analyses in this study were performed independently for groups formed with respect to the time between the onset of symptoms and collection of the sample (A, B, C, and D).” |
||
Comments 8: Line 315, it is better to use [spike] than [S]. |
||
Response 8: After reviewing another comment, the mentioned line was excluded from the manuscript, therefore the change was no longer necessary. |
||
Comments 9: Line 315, [higher transmission] or [higher replication]? |
||
Response 9: After reviewing the paragraph, the mentioned line was excluded from the manuscript, therefore the clarification was no longer necessary. |
||
Comments 10: If the results regarding anti-SARS-CoV2 therapy are available in these subjects, it is better to include the relationship between anti-viral effect and viral load, viral variants in this study. |
||
Response 10: None of the patients had access to direct antiviral therapy against SARS-CoV-2. During the study period, only palliative therapies were available. |
||
4. Response to Comments on the Quality of English Language |
||
Point 1: English editing is suggested: [a] The paragraph from lines 47 to 54 is similar to that from 55 to 60; [b] Lines from 225 to 228; [c] Lines from 346 to 349. |
||
Response 1: a) One of the paragraphs was removed from the manuscript. b) The paragraph that started at line 225 was rewritten: “As shown in Figure 4, older adults had a lower viral load compared to the other age groups (Group A: 0-9: -5.0734, 10-19: -5.6155; 20-59: -5.3863; ≥60: -4.6224; p <0.05) Group B (0-9: -4.7212, 10-19: -4.8003; 20-59: -4.4685; ≥60: -3.9526; p< 0.05), and the differences were significant only in comparison with the groups of 10-19 and 20-59 years, and only during the first four days after the onset of symptoms.” c) The text that was originally written on line 346 has been rewritten: “However, unlike our results, Tsukagoshi and collaborators [30] found that the viral load in deceased patients was significantly higher; however, we must consider that in their study, the sample was composed of only 286 individuals. Similarly, Liu et al also concluded that patients with severe COVID-19 tend to have a high viral load and a long virus shedding period [31].” |

Round 2
Reviewer 2 Report
Comments and Suggestions for Authors
The issues I raised previously have been addressed in this revised manuscript.